# Comparison of Incisional Hernia Rates Between General and Gynecological Surgery Procedures

**DOI:** 10.3390/medicina61030435

**Published:** 2025-02-28

**Authors:** Krista Spear, Daniel L. Davenport, Lance Butler, Margaret Plymale, John Scott Roth

**Affiliations:** 1College of Medicine, University of Kentucky, Lexington, KY 40536, USA; krista.spear@uky.edu; 2Division of Healthcare Outcomes and Optimal Patient Services, Department of Surgery, University of Kentucky, Lexington, KY 40536, USA; daniel.davenport@uky.edu (D.L.D.); lance.butler@uky.edu (L.B.); 3Division of General, Endocrine, and Metabolic Surgery, Department of Surgery, University of Kentucky, Lexington, KY 40536, USA; mplym0@uky.edu

**Keywords:** hernia, incisional hernia, hernia rate comparison, gynecological surgery, general surgery

## Abstract

*Background and Objectives*: Incisional hernias are a common and costly complication of surgery, occurring in up to 20% of midline incisions within 3 years of initial operation. Risk factors for incisional hernia include incision site, fascial closure technique, body mass index (BMI), surgical site infections, and gastrointestinal surgery. Limited studies have compared procedural type as a risk factor for hernia formation. The goal of this study was to examine incisional hernia rates among general surgical and gynecologic procedures. *Materials and Methods*: We queried our Research Data Warehouse for inpatients who had undergone common open abdominal surgeries between January 2012 and December 2022. Patients’ index operations were identified based upon Current Procedural Terminology (CPT) codes and presence of a postoperative incisional hernia was determined by occurrence of an incisional hernia ICD10 diagnosis code more than 2 weeks postoperatively. The main study outcome was time to incisional hernia diagnosis. *Results*: A total of 4447 patients were identified. Postoperatively, 241 (5.4%) patients were diagnosed with incisional hernias. Hernia rates at 1, 3 and 5 years were 3% (SE 0.003), 6% (0.004) and 8% (0.005), respectively. Patients undergoing exploratory laparotomy (hazard ratio 3.9, *p* < 0.001), bowel resection (HR 5.5, *p* < 0.001), and primary hernia repair (HR 13.0, *p* < 0.001) were found to have significantly increased risk for incisional hernia development compared to those undergoing hysterectomy, following adjustment for comorbid risks, age, sex, and BMI. *Conclusions*: Exploratory laparotomy, bowel resection, and primary ventral hernia repair are associated with a higher incidence of incisional hernia relative to gynecologic procedures. This relatively unstudied comparison warrants further investigation.

## 1. Introduction

Incisional hernias (IH) are one of the most common complications following midline laparotomy, with reported incidence ranging between 2% and 22% [1,2,3]. Not only can incisional hernias lead to significant discomfort for patients, but they also place a significant financial burden on the patients and the health care system overall. Alli, Zhang, and Telem demonstrated that the development of IH increases the cost of care by 97–310% over the three years following the index operation, with mean expenditures being over USD 20,000 higher than for patients who do not develop IH [4]. Furthermore, the patients who acquire IH in the first year following surgery experience a significantly higher treatment cost than those who develop IH more than two years after the operation [4]. Increased cost of care associated with IH has been noted in multiple studies, with average readmission costs and overall combined cost of care being significantly higher for patients with hernia than those without [5]. This extends to health systems as well, with one large, academic medical system reporting costs exceeding USD 17.5 million over seven years for the management of hernia and the related complications [5].

Gynecological operation has been reported to be a unique risk factor for incisional hernia [6]. IH incidence after gynecologic surgery has been reported to vary between 2% and 17% with similar risk factors identified as those in gastrointestinal surgery [7]. Previous research has found that the overall incidence of IH following Pfannenstiel incisions and off-midline incisions generally is significantly less than with a midline laparotomy [8]. While Pfannenstiel incisions are commonly used for procedures such as cesarean sections, they are often not appropriate, particularly among gynecological oncology procedures where more broad exposure is needed [9]. Limited studies compare gynecological and general surgical procedures in terms of IH incidence, with hypotheses as to rationale for IH formation amongst gynecologic procedures to include incision type and technique [7,10,11]. Meanwhile, gastrointestinal surgery hernia rates are likely related to midline incisions alone [12].

The aim of this study was to examine incisional hernia rates in patients who have undergone open general surgical procedures compared with gynecological procedures to determine the impact of specialty intervention upon incisional hernia rates among procedures typically requiring a midline incision.

## 2. Materials and Methods

The University of Kentucky Medical Institutional Review Board (IRB) reviewed and approved this study and allowed a waiver of informed consent. Common surgical procedures that use a midline incision when performed open were identified and reviewed including exploratory laparotomy, small bowel resection, large bowel resection, primary ventral hernia repair, and hysterectomy. Patients who had undergone a prior ventral hernia repair and therefore having recurrent ventral hernia repair were excluded. With IRB approval, we queried our Research Data Warehouse via the Center for Clinical and Translational Science at the University of Kentucky for inpatients who had undergone these common open abdominal procedures between January 2012 and December 2022. The procedures were identified by their corresponding Current Procedural Terminology (CPT) codes (Table 1). The date of the first diagnosis of incisional hernia in the EMR occurring at least two weeks following the index surgery was identified using the corresponding ICD 10 code. Patient exclusions included those without post-surgery encounters in the database at least two weeks after the initial operation, patients with more than one qualifying surgery hospitalization, and patients with prior or concurrent diagnosis of ventral hernia. Patient identifiers were removed and patient sex, age, BMI, Charlson Comorbidity Index, as well as the following comorbidities were also extracted: abdominal aortic aneurysm, diverticular disease, chronic obstructive pulmonary disease, smoking status, and hypertension.

Time to incisional hernia diagnosis was analyzed using the Kaplan–Meier method with log-rank tests performed to detect differences in groups. Multivariable Cox regression was used to assess group differences with adjustment for demographic and comorbid risk factors. All statistical calculations were performed using SPSS© statistical software version 28 (IBM© Corp., Armonk, NY, USA). Significance was set at *p* < 0.05 for all analyses.

## 3. Results

A total of 4447 patients were identified to be included in the analyses. Patients’ mean age at the time of the index procedure was 53.37 years (±SD 14.48), with the majority (75.2%) of patients being female. Of the identified patients, the index procedures were distributed as follows: 5.4% underwent a primary ventral hernia repair, 9.6% underwent an exploratory laparotomy, 34.9% underwent a bowel resection (including large or small bowel), and 50.1% underwent a hysterectomy (Table 2). The specific procedures were compiled into one of these four overarching procedure categories as shown in Table 1, and specific procedure counts and corresponding codes can be found in Table 3.

A total of 241 (5.4%) patients were diagnosed with incisional hernia. Overall herniation rates at 1, 3 and 5 years were 3% (SE 0.003), 6% (0.004), and 8% (0.005), respectively. Of the comorbid factors analyzed in this study, 50.5% of patients were hypertensive, defined by use of hypertensive medications, 7.2% had a diagnosis of COPD, 27.2% had used tobacco within the past 30 days, 0.7% had an identified abdominal aortic aneurysm, and 7.3% had a diagnosis of diverticular disease (Table 4). Comorbid factors found to reduce time to hernia diagnosis during Kaplan–Meier analysis included sex (*p* < 0.001), COPD (*p* = 0.012), hypertension (*p* = 0.045), and diverticular disease (*p* < 0.001) (Table 5). Diverticular disease was a strong predictor of decreased time to incisional hernia diagnosis with a hazard ratio of 1.47 (95% CI 1.04–2.08, *p* = 0.30). While found to be a significant risk factor within bivariate Kaplan–Meier analysis, as an independent factor, COPD was not found to be a statistically significant predictor of time to incisional hernia diagnosis with a hazard ratio of 1.45 (95% CI 0.94–2.23, *p* = 0.092). The Charlson Comorbidity Index was found to have diminishing significance as an independent predictor of time to incisional hernia diagnosis after open abdominal surgery as more comorbidities were identified in patients. For patients with 1–2 comorbidities present, the hazard ratio was 1.73 (95% CI 1.12–2.66, *p* = 0.014), while patients who had 3–4 comorbidities were found to have a hazard ratio of 1.52 (95% 0.96–2.39, *p* = 0.073). Patients with 5–7 comorbidities had a hazard ratio of 1.34 (95% CI 0.83–2.17, *p* = 0.226), and those with 8 or more had a hazard ratio of 1.26 (95% CI 0.78–2.01, *p* = 0.341). The average BMI of patients was 31.79 (±SD 10.17). Patient BMI greater than or equal to 25 was found to be statistically significant as an independent predictor of time to incisional hernia diagnosis (*p* < 0.01). Analysis of independent risk factors can be seen in Table 6.

Herniation hazards varied significantly by procedure type after adjustment for comorbid risks, age, sex, and BMI (Figure 1). Patients undergoing exploratory laparotomy (hazard ratio 3.9, *p* < 0.001), bowel resection (HR 5.5, *p* < 0.001), and primary hernia repair (HR 13.0, *p* < 0.001) had increased risk for incisional hernia development compared to hysterectomy patients, with adjustment for comorbid and demographic risk factors.

## 4. Discussion

The high incidence and financial burden of incisional hernia has led to substantial research aiming to identify risk factors for IH that may be modifiable and that could identify higher risk individuals for closer monitoring. These risk factors are often stratified into those related to the patient, those due to the disease process, and those affected by the surgeon. Patient factors can be subdivided into those that are modifiable and those that are non-modifiable [4]. The former include many that affect how the body heals postoperatively via metabolic effects or physical stress, including hyperglycemic or diabetic status, obesity and body mass index (BMI), smoking status, immunosuppressant or steroid use, and malnutrition [13]. This is often the subset of factors that physicians attempt to optimize preoperatively. The non-modifiable risk factors include prior laparotomies, chronic infections, malignancy, sex, and age. Disease factors that impact the incidence of incisional hernia include emergent vs. elective surgery status, preoperative peritonitis, history of radiation to the abdominal cavity, and postoperative surgical site infection [14]. Technical factors associated with IH incidence include the incision type, the choice of the suture material, wound closure technique, and the expertise of the performing surgeon. Midline incisions, especially those in the lower midline, are more likely to develop hernia than a transverse incision, while both are more likely than a laparoscopic incision to develop hernia [15]. Slowly absorbable sutures have been identified as the preferable suture material, as they have been shown in multiple meta-analyses to decrease the risk of surgical site infection and to decrease incidence of hernia [15]. More importantly still is the technique used to close the wound, with the small bite technique, 5 mm bites with 5 mm advance, being found to be more effective than traditional large bites at preventing IH [16]. While this is not an exhaustive list of risk factors that impact the incidence of incisional hernia, it does include those most studied at this time.

This large sample study performed at a single institution demonstrated that patients undergoing general surgical operations with midline incisions have decreased time to formation of incisional hernia than patients undergoing gynecological procedures with midline incisions. This study is important as it is the first to be of large scale and to directly compare general surgical procedures and gynecological procedures. It contributes to a group of mixed findings on the topic. Determining if the results are replicable at other large institutions and among different populations will be important. In two previous studies that compared the two fields, fewer than 50 cases were included in each, and both studies found that IH was more common after gynecological operation [7,12]. It was suggested that it may be due to use of the infra-umbilical midline incision where there is an absence of the posterior rectus sheath, which in combination with a weak abdominal wall due to histories of multiple child births, could have put these patients at an increased risk for hernia.

Another possible explanation is the difference in education regarding abdominal wall fascial closures between general surgeons and gynecologists. It was previously thought that larger stitches, those placed at least 10 mm from the wound edge, would produce a stronger wound. An experimental study conducted by Cengiz, Blomquist, and Israelsson countered that idea, finding that long stitches compressed and cut through the tissue contained in the stitch, increasing the amount of necrotic tissue compared to smaller stitches, placed 3–6 mm from the wound edge [16]. Prior to this, Jenkins proposed that an incision be closed with a suture to wound length ratio of at least 4, as when it is less than 4, the risk of herniation is three times higher [17]. More recently, the STITCH trial demonstrated that the small bite technique, 5 mm bites with 5 mm advance, to be more effective than traditional large bites at preventing IH [16]. One study found that surgery residents were significantly more likely than Obstetrics and Gynecology (OB/Gyn) residents to know the correct suture-to-wound-length ratio and be familiar with this literature on abdominal wall closure [18]. This was consistent with a practical evaluation as well, with surgery residents taking significantly smaller bites of fascia with shorter advances consistent with the short bite techniques than their OB/Gyn resident colleagues. This discrepancy in knowledge would imply that general surgeons would be more likely to utilize the proper fascial closure technique than gynecologists. However, this would have led us to opposite findings, in which general surgical procedures would have had a longer time to hernia diagnosis than gynecological procedures. Given that this was not found, it leads us to wonder if there is another explanation for this discrepancy, such as a difference in planned vs. emergent status or possible mislabeling of incision type. To delve into this further, closer analysis of operative reports would be needed.

We were also able to further analyze comorbid factors that impact time to hernia diagnosis. We determined that gender, hypertension, COPD, and diverticular disease all were significant risk factors (*p* < 0.05) that contribute to the herniation hazard. Interestingly, BMI was not found to significantly impact the herniation hazard (*p* = 0.071) but was found to be an independent predictor of time to incisional hernia diagnosis when the BMI was greater than 25 (*p* < 0.001). Though we expected it to have an impact on herniation hazard, BMI as an independent predictor is consistent with previous findings. Patients being of older age had previously been identified as a predisposing risk factor for IH, in this study it was not found to impact the herniation hazard (*p* = 0.21). This may be due to the lower average age of the patients analyzed in the study, as some research has shown the risk for IH increases in the elderly primarily over the age of 70 [19]. Interestingly, sex of the patient was found to be a significant factor impacting the herniation hazard (*p* < 0.001) but was not found to be an independent predictor of time to incisional hernia diagnosis (*p* = 0.113). This is contrary to previous findings that show male sex is a risk factor for herniation. Diverticular disease, though found in a small group of patients in this study, was found to be a significant factor that impacts the herniation hazard (*p* < 0.001) and is a significant independent predictor of time to incisional hernia diagnosis (*p* = 0.03). This is consistent with previous research which suggested that diverticular disease is linked to genes that affect connective tissue formation associated with the development of abdominal wall hernias [20]. Nicotine use was not found to be a significant factor that affects herniation hazard (*p* = 0.247). This may be due to a lack of clear documentation of smoking or tobacco status in the charts of the patients we analyzed. Abdominal aortic aneurysm was not found to affect the herniation hazard (*p* = 0.326) in this study, potentially due to a very low incidence of the comorbidity in the analyzed population, only documented in 33 patients. The Charlson Comorbidity Index is a tool that predicts 10-year survival in patients with multiple comorbidities, analyzing the presence of 17 possible comorbid risk factors. In this study, it was found to impact time to herniation with decreasing utility as more comorbid factors were present, only significantly impacting time to herniation when there were 1–2 comorbid factors identified (*p* = 0.014). Very few studies have used this index as a predictor for incisional hernia development; however, one analyzing hernia after radical prostatectomy did find >2 comorbid factors to be significant in increasing risk for herniation repair [21]. Overall, many of the findings observed were consistent with previous risk factors that have been identified, though some were found to be less significant than previously suggested.

This study has many limitations. First, this was a retrospective study, so we are unable to draw definitive conclusions regarding cause and effect. CPT codes were selected based upon the likelihood of each procedure to be performed with a midline incision. However, the closure technique, length of incision, and location of incision were not able to be captured from the database. These technical details may be relevant and impactful, but prospective studies are required to gather this level of detail. This study was also conducted at a single, large academic center with analysis of mostly women which is not truly representative of the population that incisional hernia affects. However, this study’s large sample size increases the likelihood of generalizable findings. Furthermore, the evaluated procedures may have been performed in different population groups. Given that bowel resections and exploratory laparotomies are sometimes performed in emergent or urgent situations, this could impact the risk of IH by as much as 42–50% [16]. On the contrary, hysterectomy is typically an elective procedure. This could partially account for the reduced rate of IH in gynecologic procedures.

## 5. Conclusions

In this study, we observed that individuals who underwent general surgical procedures, such as exploratory laparotomy, bowel resection, and primary hernia repair, were at increased risk for incisional hernia diagnosis when compared to those undergoing a hysterectomy. These findings, at the scale in which this study was performed, are novel and contrary to previous studies that endorse increased risk of hernia with gynecological operations. Potential variables associated with increased risk for incisional hernia development for patients who underwent exploratory laparotomy, bowel resection, and primary ventral hernia repair compared to those undergoing hysterectomy include emergent vs. elective status, patient age, patient gender, preoperative infection, and method of closure, among other factors. This topic warrants further investigation to determine replicability and the possible etiology of the differences. Awareness of the impact of procedural type upon incisional hernia rates has potential ramifications in future strategies for hernia prevention.

## Figures and Tables

**Figure 1 medicina-61-00435-f001:**
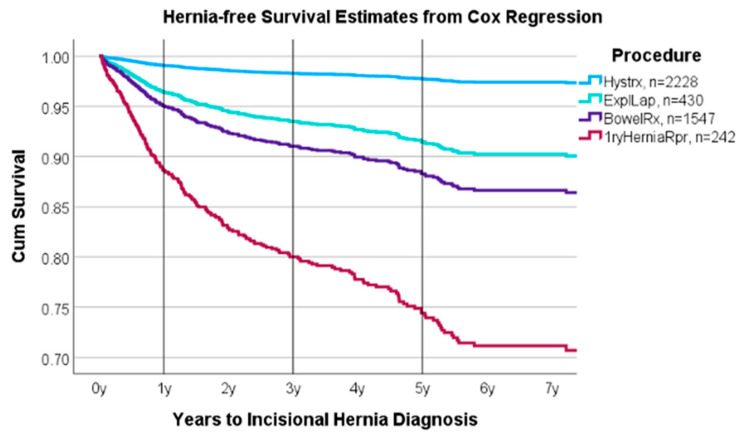
Cox regression estimates of time to incisional hernia by index procedure type following adjustment for comorbid factors. Exploratory laparotomy (ExpLap; green line) (hazard ratio 3.9, *p* < 0.001), bowel resection (BowelRx; purple line) (HR 5.5, *p* < 0.001), and primary hernia repair (1ryHerniaRpr; red line) (HR 13.0, *p* < 0.001) were all found to have significantly increased risk for incisional hernia when compared to hysterectomy (Hystrx; blue line) after adjustment for comorbid factors.

**Table 1 medicina-61-00435-t001:** CPT Codes Corresponding to Analyzed Procedures.

Procedures Analyzed	Corresponding CPT Codes
Primary Ventral Hernia Repair	49591, 49592, 49593, 49594, 49595, 49596
Exploratory Laparotomy	49000, 49002
Small or Large Bowel Resection	44140, 44141, 44144, 44143, 44150, 44155, 44157, 44158, 44160, 44120, 44121, 44125-8, 44130
Hysterectomy	58150, 58200, 58210

**Table 2 medicina-61-00435-t002:** Procedure Type Distribution in Analyzed Population.

Procedure Type	Count (% of Analyzed)
Primary Ventral Hernia Repair	242 (5.4)
Exploratory Laparotomy	430 (9.6)
Bowel Resection	1547 (34.9)
Hysterectomy	2228 (50.1)

**Table 3 medicina-61-00435-t003:** Procedure Type Based on CPT Code and Percentage of Total Analyzed Procedures.

Primary Procedure	Count (Percent)
44120-ENTERECTOMY W/ANASTOMOSIS	554 (12.5%)
44121-ENTERETOMY EACH ADDITIONAL RESECT W/ANASTO	25 (0.6%)
44130-ENTEROENTEROSTOMY ANASTOM W/WO ENTEROSTOMY	30 (0.7%)
44140-COLECTOMY, PARTIAL W ANASTOMOSIS	342 (7.7%)
44141-COLECTOMY PARTIAL W/CECOSTOMY OR COLOSTOMY	66 (1.5%)
44143-COLECTOMY, END COLOSTOMY	224 (5.0%)
44144-COLECTOMY W/RESECTION COLOSTOMY	66 (1.5%)
44150-COLECTOMY, TOTAL W/ILEO	71 (1.6%)
44155-COLECTOMY W/PROCTEC&ILEO	15 (0.3%)
44157-COLECTOMY TOT W/ILEONAL ANASTOMIS	3 (0.1%)
44158-COLECTOMY W/ILEONAL ANAST INCL LOOP ILEOST	9 (0.2%)
44160-COLECTOMY, REM TER ILEUM	142 (3.2%)
49000-EXPLORATORY LAPAROTOMY	304 (6.8%)
49002-REOPEN LAP INC FOR EXPLR	126 (2.8%)
49560-VENTRAL HERNIA REPAIR	242 (5.4%)
58150-TOTAL HYSTERECTOMY	1777 (40.0%)
58200-TOT HYSTERECT W/PAR VAG W/LYMPH NODE BX	311 (7.0%)
58210-RADICAL HYSTEREC WITH LYMPHADENECTOMY	140 (3.1%)
Total	4447 (100%)

**Table 4 medicina-61-00435-t004:** Comorbid Factors Observed in Analyzed Population.

Variable	Count (Percent)
Number of Total Patients	4447
Female Sex	3343 (75.2%)
Hypertensive	2246 (50.5%)
SUD/Nicotine	899 (20.2%)
Tobacco Use 30-day	1211 (27.2%)
Abdominal Aortic Aneurysm	33 (0.7%)
Diverticular Disease	325 (7.3%)
COPD	320 (7.2%)

**Table 5 medicina-61-00435-t005:** Comorbid Factors Bivariate Herniation Hazards Based on Kaplan–Meier Analysis.

Variable	*p*-Value
Gender	<0.001
Hypertensive	0.045
SUD/Nicotine	0.247
Abdominal Aortic Aneurysm	0.326
COPD	0.012
BMI Group	0.071
Age Decade	0.208
Charlson Index Quintile	0.086
Diverticular Disease	<0.001

**Table 6 medicina-61-00435-t006:** Independent Predictors of Time to Incisional Hernia Diagnosis after Open Abdominal Surgery.

Variable	HR (95% CI)	*p*-Value
Female vs. Male	1.25 (0.95–1.66)	0.113
COPD	1.45 (0.94–2.23)	0.092
BMI vs. ≤25 kg/m^2^	0.00 (0.00–0.00)	<0.001
25.01–30	1.75 (1.18–2.58)	0.005
30.01–35	2.00 (1.32–3.03)	0.001
35.01–40	2.09 (1.31–3.33)	0.002
>40	3.11 (2.02–4.81)	<0.001
Charlson Index vs. = 0	0.00 (0.00–0.00)	0.117
1–2	1.73 (1.12–2.66)	0.014
3–4	1.52 (0.96–2.39)	0.073
5–7	1.34 (0.83–2.17)	0.226
≥8	1.26 (0.78–2.01)	0.341
Diagnosis of Diverticular Disease	1.47 (1.04–2.08)	0.030
CPT Codes Grouped by 1st 3 digits compared to 441XX vs. 441		
490XX (exploratory laparotomy)	6.90 (3.16–15.09)	<0.001
495XX (ventral hernia repair)	4.99 (2.10–11.84)	<0.001
581XX (total hysterectomy)	16.16 (7.23–36.12)	<0.001
582XX (total hysterectomy with other)	1.37 (0.60–3.12)	0.460

## Data Availability

The dataset presented in this article is not readily available because of an ongoing study and ethical considerations. Requests to access the datasets should be directed to the corresponding author.

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
