# Peer review of "Comparison of Incisional Hernia Rates Between General and Gynecological Surgery Procedures"

_medicina, 2025, doi:10.3390/medicina61030435_

Round 1

Reviewer 1 Report

Comments and Suggestions for Authors

This study is a large retrospective cohort examining midline incisional hernia rates between gen surg and OBGYN procedure. It shows that hernia rates are higher in common gen surg procedures than hysterectomies. 

Some questions and suggestions.

Please provide a justification as to the variables picked for cox regression. For example, why was diverticular disease picked as a comorbidity? I would think diabetes would be a much more relevant variable.

In the intro you state that patients were excluded with a prior ventral hernia, yet ventral hernia repair is listed as one of the procedures analyzed. Is this a contradiction?

Define the CPTs in Table 6.

Including a figure in the the discussion postulating differences in hernia rates between gen surg and obgyn would be helping in conceptualizng and organizing the various postulates proposed.

Author Response

1. Summary

2. Point-by-point response to Comments and Suggestions for Authors
- Reviewer #1

Comment 1: Please provide a justification as to the variables picked for cox regression. For example, why was diverticular disease picked as a comorbidity? I would think diabetes would be a much more relevant variable.

Response 1: Diabetes is prominently featured in the Charlson Comorbidity Index (one of 19 variables). We chose the other to be those not in Charlson. A backwards stepwise approach including Charlson and those listed was used to narrow the final list to those significantly predictive of recurrence, p< .10.

Comment 2: In the intro you state that patients were excluded with a prior ventral hernia, yet ventral hernia repair is listed as one of the procedures analyzed. Is this a contradiction?

Response 2: Thank you for pointing out that this is an area of confusion. We excluded cases of recurrent ventral hernia repair. We have added text to the first paragraph of the Materials and Methods section to clarify.

Comment 3: Define the CPTs in Table 6.

Response 3:  We have revised Table 6 for clarity. We combined the codes based on the first 3 digits of the CPT code so that the groups get combined are larger but still accurate down to the organ.

Comment 4: Including a figure in the discussion postulating differences in hernia rates between gen surg and obgyn would be helping in conceptualizng and organizing the various postulates proposed

Response 4: After review, rather than add another table or figure to the manuscript, the authors opted to add to the Conclusions section the following text: Potential variables associated with increased hazard for incisional hernia development for patients who underwent exploratory laparotomy, bowel resection and primary ventral hernia repair compared to those undergoing hysterectomy include emergent vs. elective status, patient age, patient gender, preoperative infection, method of closure, among other factors.

Reviewer 2 Report

Comments and Suggestions for Authors

I have read with great interest the article entitled “Comparison of Incisional Hernia Rates between General and Gynecological Surgery Procedures” in order to evaluate its publication in the journal Medicina

This is a retrospective study that compares the rate of incisional hernia between surgical procedures performed in the specialty of General Surgery Vs Gynecology.

An important aspect of the study is the low rate of incisional hernia (5.4% overall, 3% at the 1st year, 6% at the 2nd and 8% at the 5th), what do you think is due to this? How do you justify it?

In my opinion, including primary ventral hernia as another procedure is a mistake since it implies a different type of problem.

The introduction is too broad and includes aspects that correspond more to the discussion and therefore should not be in this section.

It is not clear to me how the follow-up and inclusion of patients with incisional hernia criteria is carried out.

Many risk factors are analyzed, but I believe there are many others, of great importance, that are not analyzed, such as the CDC classification of the wound, the size of the laparotomy, its location, postoperative complications, whether it is performed in an emergency or not...

At no time is the laparotomy closure technique described; such an important technical aspect cannot go unnoticed.

Minor errors:

- CPT abbreviation used in the abstract without justifying its meaning.

- Meaning of CPT3 in table 6.

- High number of exploratory laparotomies (304 cases).

Author Response

  1. Point-by-point response to Comments and Suggestions for Authors
- Reviewer #2

This is a retrospective study that compares the rate of incisional hernia between surgical procedures performed in the specialty of General Surgery Vs Gynecology.

Comment 1: An important aspect of the study is the low rate of incisional hernia (5.4% overall, 3% at the 1st year, 6% at the 2nd and 8% at the 5th), what do you think is due to this? How do you justify it?

Response 1: The scope of this study does not allow us to fully understand the incisional hernia rate. A limitation of this study is its retrospective nature and likely there were incisional hernias that were not captured for a variety of reasons including follow up at another facility the data from which we would not have.

Comment 2: In my opinion, including primary ventral hernia as another procedure is a mistake since it implies a different type of problem.

Response 2: We opted for cases that included a laparotomy. We chose to include open repair of primary ventral hernia due to the volume of cases. Additionally, CPT codes were selected based upon the likelihood of each procedure to be performed with a midline incision.  

Comment 3: The introduction is too broad and includes aspects that correspond more to the discussion and therefore should not be in this section.

Response 3: Thank you for this comment. The authors agree with you, and we have readjusted the introduction and discussion sections of the manuscript.

Comment 4: It is not clear to me how the follow-up and inclusion of patients with incisional hernia criteria is carried out.

Response 4: Incisional hernia in these cases was determined by identifying the date of the first diagnosis of incisional hernia in the electronic health record occurring at least two weeks following the index surgery. Incisional hernia was identified using the corresponding ICD 10¬ code.

Comment 5: Many risk factors are analyzed, but I believe there are many others, of great importance, that are not analyzed, such as the CDC classification of the wound, the size of the laparotomy, its location, postoperative complications, whether it is performed in an emergency or not.

Response 5: We appreciate your insight into this limitation of our manuscript. While we agree with you as to the importance of the mentioned risk factors, we were not able in this early study to include these data points.

Comment 6: At no time is the laparotomy closure technique described; such an important technical aspect cannot go unnoticed.

Response 6: The authors agree as to the high level of importance of closure technique; however as mentioned in the discussion section of the manuscript, operative reports would have had to have been reviewed to obtain this information, and this study does not include operative report review.

Comment 7 :  CPT abbreviation used in the abstract without justifying its meaning.

- Meaning of CPT3 in table 6.

- High number of exploratory laparotomies (304 cases).

Response 7:

The authors appreciate these comments, and we have added Current Procedural Terminology prior to using CPT in the abstract. We have revised Table 6 to provide clarity as to the CPT codes used. Concerning the high number of exploratory laparotomies, our facility includes a high volume Level I Trauma Center, and these cases are included in the dataset for this study.

Round 2

Reviewer 2 Report

Comments and Suggestions for Authors Minor changes have been made to the manuscript that do not substantially change it, so I stand by the decision made previously.